# Knowledge-Aware Artifact Image Synthesis with LLM-Enhanced Prompting and Multi-Source Supervision

## ABSTRACT

Ancient artifacts are an important medium for cultural preservation and restoration. However, many physical copies of artifacts are either damaged or lost, leaving a blank space in archaeological and historical studies that calls for techniques to re-visualize these artifacts. Despite the significant advancements in open-domain text-to-image synthesis, existing approaches fail to capture the important domain knowledge presented in the textual descriptions of artifacts, resulting in errors in recreated images such as incorrect shapes and patterns. In this paper, we propose a novel knowledge-aware artifact image synthesis approach that brings lost historical objects accurately into their visual forms. We use a pretrained diffusion model as backbone and introduce three key techniques to enhance the text-to-image generation framework: **1)** we construct prompts with explicit archaeological knowledge elicited from large language models (LLMs); **2)** we incorporate additional textual guidance to correlated historical expertise in a contrastive manner; **3)** we introduce further visual-semantic constraints on edge and perceptual features that enable our model to learn more intricate visual details of the artifacts. Compared to existing approaches, our proposed model produces higher-quality artifact images that align better with the implicit details and historical knowledge contained within written documents, thus achieving significant improvements both across automatic metrics and in human evaluation. Our code and data will be made publicly available.

## CCS CONCEPTS

• **Computing methodologies** → **Computer vision**; • **Applied computing** → **Archaeology**.

## KEYWORDS

Ancient Artifact Visualization, Text-to-Image Synthesis, Diffusion Models, Multi-Source Supervision, Large Language Models

## 1 INTRODUCTION

Ancient artifacts are crucial for cultural preservation, as they represent tangible evidence of the past, offering insights into history. In recent years, innovative artifact-related projects have emerged, including the restoration of degraded character images [33], the generation of captions for ancient artwork [32], and the deciphering of oracle bone inscriptions [5]. These works have opened up

**Unpublished working draft. Not for distribution.**

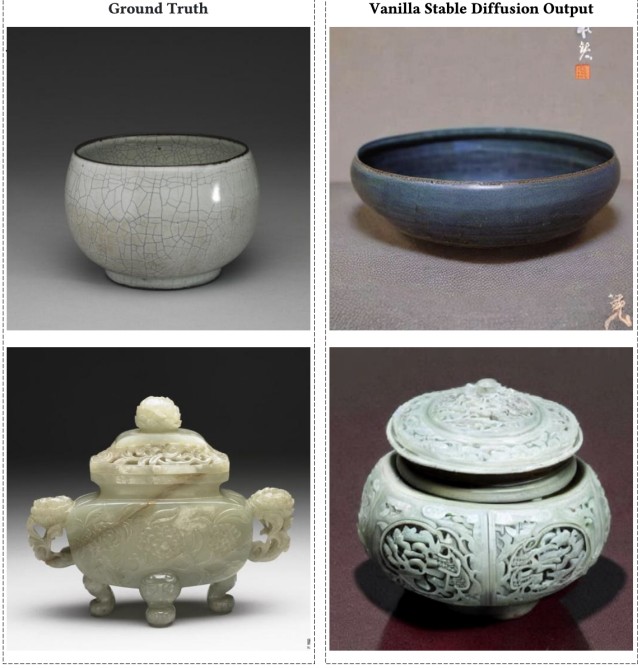

**Figure 1: Images of artifacts generated by a vanilla diffusion model. The shape, color, pattern, and material-look differ greatly from the ground truth.**

new avenues for researchers to study artifacts and gain insights into the past. Despite these advancements, there are still many areas to be explored in artifact-related tasks, one of which is to recreate visual images of artifacts from text descriptions, as many physical copies of artifacts are often damaged or lost, leaving only textual records behind. This task could prove immensely invaluable to historical studies and cultural preservation because it provides historians with new visual angles to study the past and enables people to connect with their cultural heritage.

One line of techniques that has shown potential to aid in the recreation of visual images of ancient artifacts is text-to-image synthesis. This task has been a popular area of research, especially in recent years with the introduction of diffusion models [14, 22, 27, 36, 39, 49] that have demonstrated significant capabilities in generating photo-realistic images based on a given text prompt in open-domain problems [10, 23, 26, 27, 30]. However, in the specialized area of archaeological studies, where data is often limited and domain knowledge is required, vanilla diffusion models struggle to produce promising results even with finetuning, as shown in Figure 1. The generated images often display errors in shape, patterns, and details that fail to match the implicit knowledge

in the textual information and the underlying historical context of the target artifact.

We have identified a key cause for this problem to be the lack of knowledge supervision during the generating process, which can be attributed to two main aspects. 1). Current text prompts may not be infused with domain-specific knowledge from the archaeological and historical fields, leading to noisiness and the lack of well-presented knowledge information in the text prompt. 2). The text and visual modules in the vanilla diffusion models [14, 27, 35, 37, 38] may be unable to capture domain-specific knowledge under the standard training pipeline, resulting in the absence of detailed textual and visual signals of ancient artifacts in the generation process.

To address these challenges, we propose our knowledge-aware artifact image synthesis approach with a pretrained Chinese Stable Diffusion model [27, 51] as our backbone. Our method can generate visualizations of lost artifacts that well align with the underlying domain knowledge presented in their textual records. Specifically:

1). To address the issue of noisiness and lack of well-presented knowledge information in the text prompt, we propose to use Large Language Models (LLMs) to enhance our text prompts in two ways: for one, we use LLMs to extract the core and meaningful information in the given text prompt and reorganize them in a more structured way to explicitly present the current knowledge information; for another, we use LLMs as an external knowledge base to retrieve relevant archaeological knowledge information and augment them in the restructured text prompt.

2). To address the lack of both textual and visual knowledge supervision in the generation process, we introduce additional supervision in both modalities. Firstly, we introduce a contrastive training paradigm that enables the text encoder to make the textual representation of the artifact more in line with their archaeological knowledge. Secondly, we apply stricter visual constraints using edge loss [31] and perceptual loss [15] to make the final visual output align with the visual domain knowledge of ancient artifacts.

Both quantitative experiments and a user study involving human experts demonstrate that our knowledge-aware artifact image synthesis approach significantly outperforms existing text-to-image models and greatly improves the generation quality of historical artifacts.

Overall, our main contributions can be summarized as follows:

- To our best knowledge, we are the first to explore the text-to-image synthesis task in archaeology as an attempt to **visually recreate lost historical artifacts**, thus aiding archaeologists in gaining deeper insights into our past cultural treasury.
- We propose to use LLMs as both an information extractor and external knowledge base to **elicit archaeological knowledge** that explicitly aids better prompt construction in the specialized domain requiring high historical precision.
- We introduce **additional multimodal supervisions** to enable our model to learn textual representations and visual features that better align with archaeological knowledge and historical context, thus improving the current finetuning paradigm of diffusion models.

| Li Ge Ding | Square ding cauldron with animal-mask pattern |
|---|---|
| It has upward sloping ears, angular rim, rounded body, columnar legs, and belly with two raised bands and a circular protrusion in the middle and  inscriptions on the inside walls. | Fang Ding, with erect ears, a flat bottom, feet, and angular ridges. The belly is decorated with animal faces, and the feet are adorned with geometric patterns. |

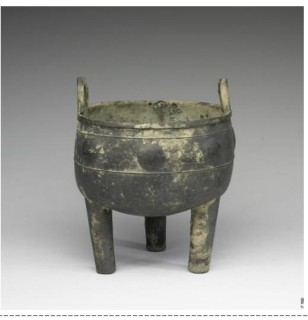 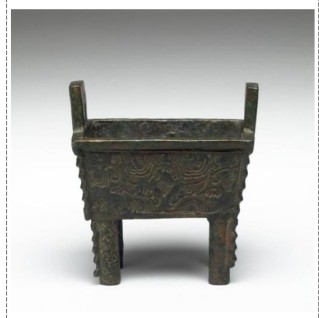

**Figure 2: Raw artifact descriptions fail to depict the artifact with sufficient archaeological information, such as their artifact-"type" which determines important aspects of their visual appearance. As shown in this figure, the artifact on the left side is classified as a "Round Ding" rather than a "Square Ding", which strictly confines its shape to a round body having only three legs as opposed to four legs.**

## 2 BACKGROUND

### 2.1 Problem Statement

As mentioned in Section 1, for many artifacts, text documents are the only available source of information. Hence our task is to recreate a visual image $I_i'$ given artifact text information $T_i$. The synthesized image $I_i'$ needs not only to align with the textual meanings conveyed in $T_i$ but also to be in line with the implicit historical knowledge about the artifact. Only then will the generated image be historically correct and thus valuable to archaeological studies. Correspondingly, the training dataset $D$ comes in the form of pairs as $D = \{(T_i, I_i)\}_i^n$, where $T_i \in T$ is the available text information and $I_i \in I$ is the corresponding artifacts image. The raw text information available for an artifact - as often cataloged in museums - contains roughly four parts: the **name** or title of the artifact; the **time period** of origin; a raw **description** of the artifact (often presented in a messy way); the physical **size** of the artifact. Formulated from accessible resources of such kind, the task of our work is then to generate accurate artifact images based on these textual descriptions of historical objects.

### 2.2 Diffusion Preliminaries

To solve the task defined in the above section 2.1, we propose to build our model upon the text-conditioned Stable Diffusion pipeline [27]. Before diving deeper into our approach, we present a detailed mathematical introduction of diffusion models and Stable Diffusion in Appendix F [1]. Here, we only briefly summarize the standard training objective of a Stable Diffusion (SD) model as follows:

$$L_{SD}(\theta) := \mathbb{E}_{t, \mathcal{E}(x_0), \epsilon} ||\epsilon - \epsilon_\theta(z_t, t, w)||^2 \quad (1)$$

---

[1] As per the submission guidelines, all technical **Appendices** mentioned in the main text are submitted separately as supplementary material.

where $\epsilon$ is the applied random noise. $z_t \in \mathcal{Z}$ is the representation of a noised image in the latent space at time step $t$ and $w$ is the encoded text representation. $\epsilon_\theta$ aims to denoise the latent space. $x_0$ is the real image at timestep 0 and $\mathcal{E}$ is the latent space encoder.

## 3 OUR METHOD

Our proposed approach for knowledge-aware artifact image synthesis is built upon a pretrained Stable Diffusion model, which retains its powerful generative capability of common domains and is further finetuned to align with the specific characteristics of ancient artifacts. The generic Stable Diffusion model, even with finetuning, however, struggles to generate visually and historically accurate artifact images and shows multifaceted errors, as is demonstrated in Figure 1.

To address these issues, we propose specific modifications at three steps in the Stable Diffusion system:

1). Given source text information $T_i$ of an artifact, we pass it into an LLM (in our case, *GPT-3.5-TURBO*) with carefully designed querying message and in-context examples to obtain a clean and augmented prompt input of our diffusion model $T_i'$. (Section 3.1).

2). During training, when $T_i'$ is passed into the text encoder, we apply an additional contrastive learning module on the text encoder to align the description of an artifact with its name, which is essentially an expert summary of its description. (Section 3.2).

3). After the added noise is predicted in the training phase, we reconstruct the model-predicted image $I_i'$ and apply additional visual-semantic supervision with edge loss [31] and perceptual loss [15] to steer the generation of our model closer towards the ground-truth appearance of artifacts. (Section 3.3).

The overall framework for our approach is illustrated in Figure 3 and explained in detail in the following subsections.

## 3.1 Prompt-Construction Enhanced by LLM

We have noticed that the raw description of an artifact accessible in museum resources (as mentioned in Section 2.1) is far from ideal for prompting a text-to-image model. It is often incomplete and filled with noisy messages and fails to sufficiently depict a historical object. Other than the messiness problem, these off-the-shelf descriptions may well lack specific information about an artifact that is essential to its visual form, such as its fundamental classification (or: artifact-"type"). An example [2] of this is given in Figure 2. As per archaeological terminology, the fact that the artifact on the left side is classified as a "**Round** Ding" rather than a "**Square** Ding" confines its shape to a round body having only **three** legs as opposed to **four** legs. However, key implicit archaeological information of this kind is often missing in the raw description of the artifact, prohibiting a text-to-image model from sufficiently understanding the association between the visual appearance and the textual prompt.

To alleviate the problems of noisiness and knowledge deficiency in the original text information, we propose to utilize an LLM as

---

[2]To maintain a consistent language usage throughout the paper, we translate all Chinese text (*e.g.*, the textual descriptions of artifacts) into English via ChatGPT.

both an information extractor to retrieve the most useful information, and as an external knowledge base to complete any missing important attributes of the artifact.

Based on archaeological expertise, we have compiled a list of key attributes that are vital for effectively describing artifacts and defining their physical forms, see Table 1. Examples of these attributes are given in Table 5 in Appendix A. While the "***name***", "***time period***" and "***size***" of an artifact are usually available in museum resources, the specific "***material***", "***shape***" and "***pattern***" need to be extracted or derived from the raw description of the object. Further, as explained above, the classified "***type***" of an artifact determines certain fundamental aspects of its looks, which are specified by the generic definition of this artifact-type (*i.e.*, "***type definition***"). It requires a general knowledge of archaeology to be able to categorize an ancient object into a certain artifact-type and to define the basic appearance of this type.

Table 1: Expert attributes of artifacts that are vital to their visual appearance according to archaeological expertise. See Table 5 in Appendix A for examples of these attributes.

| Expert Attribute | Definition |
|---|---|
| *Name* | name or title of an artifact |
| *Material* | the material an artifact is made of |
| *Time Period* | time period of origin |
| *Type* | classified type of an artifact |
| *Type Definition* | general definition of artifact type |
| *Shape* | shape and structure of an artifact |
| *Pattern* | patterns/motifs on an artifact |
| *Size* | physical dimensions of an artifact |

An LLM is well-suited for fulfilling these two tasks with its ability to obtain a certain extent of world knowledge from the massive pretraining corpus [20] and to learn to perform specialized downstream tasks using the in-context learning paradigm [3]. Specifically, we use *GPT-3.5-TURBO* as our knowledge-base LLM, and the prompt for querying GPT-3.5 is designed with a similar format following self-instruct [45]. Our prompt template consists of three parts: 1). A task statement that describes to GPT-3.5 the task to be done; 2). Two in-context examples of high quality sampled from our labeled pool of 54 artifacts written by archaeology experts; 3). The target artifacts whose "*material*", "*shape*", "*pattern*", "*type*" and "*type definition*" are left blank and need to be answered by GPT-3.5. The former 3 attributes can be retrieved from the given "*description*" and the latter 2 artifact-type related features need to be fulfilled via the world knowledge of GPT-3.5 and its in-context learning from the given human-labeled examples. An example of our prompt for querying artifact information is illustrated in Figure 7 in Appendix B.

By leveraging the power of LLM as both an information extractor and external knowledge provider, we are able to collect all the key attributes of a given artifact, which are then rearranged into the prompt to our diffusion model with a [*SEP*] (implemented as a Chinese comma in our work) splitting each key feature, as shown by the example in Table 5 in Appendix A. Such input prompt thus contains enriched text information that provides well-defined archaeology-knowledge guidance. It assists the text-to-image diffusion model in

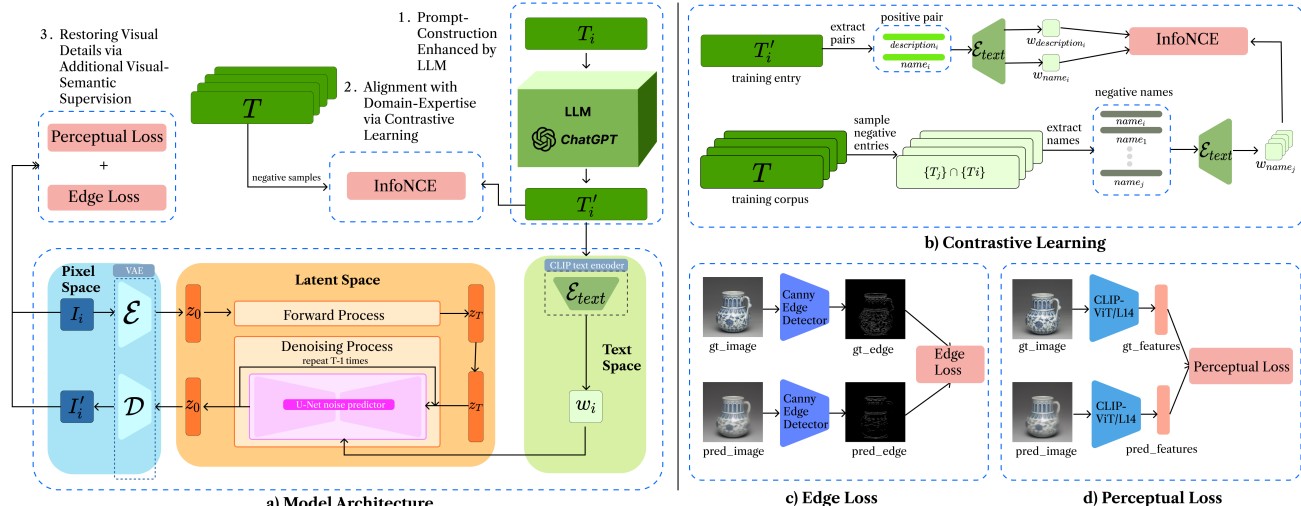

**Figure 3: Our proposed knowledge-aware approach is illustrated in a). It features a Chinese Stable Diffusion model as backbone and our proposed three key techniques labeled as follows: b) illustrates the way of performing textual contrastive learning, which is discussed in Section 3.2; c) is edge loss, and d) is perceptual loss, both of which are part of the additional visual-semantic supervision, as discussed in Section 3.3.**

synthesizing a more realistic result that better corresponds to the ground-truth artifacts.

## 3.2 Alignment with Domain-Expertise via Contrastive Learning

Another issue we identify is that the text encoder might not encode the text into a representation that reflects the underlying archaeological knowledge and thus needs further finetuning. We observe that the "***names***" of ancient artifacts are often accurate and concise summarizations of the artifact's key attributes, while the "***descriptions***" provide an extended version of the artifact's features. Given that both the names and descriptions are provided by domain experts - as written in museum sources, they reflect a high level of expertise in the field. Thus, we believe that closely aligning the names and descriptions is essential to reflect this domain knowledge.

To achieve this goal, we propose the use of contrastive learning that aims to minimize the distance between *positive* pairs consisting of matching ($[description]_i$, $[name]_i$) pairs extracted from $T_i'$, and to maximize the distance between *negative* pairs with mismatching descriptions and names.

However, we have also observed that artifacts with similar attributes (*i.e.*, similar "*description*" contents) and historical origins often share similar names, making it unintuitive to finetune the text encoder to differentiate between these similar pairs. We believe that such pairs should be close to each other in the semantic space. Therefore, we readjust our sampling strategy for negative pairs. From the perspective of historical studies, "***time period***" is one of the most determining factors in the style and appearance of an artifact, where different artifacts from different eras can be vastly different. Therefore, aiming to separate hard negatives rather

than slightly different ones, we sample our negative samples from artifact names in different eras.

In our approach, we use **InfoNCE** [42] to penalize the misalignment in the representation encoded by the text encoder. The formula for text contrastive learning can be written as:

$$L_{\text{text}} := -\mathbb{E}_X \log \frac{EXP(x_i)}{\sum_{x_j \in X} EXP(x_j)} \quad (2)$$

where $x_i = \mathcal{E}([description]_i) \cdot \mathcal{E}([name]_i)$ denotes the similarity between a pair positive sample in the text encoder's embedding space. And $X$ is the set of $N$ similarities between sampled pairs from the entire dataset, containing one positive sample $x_i$ and $N-1$ negative samples $x_j \in X$ where $i \neq j$ and $PeriodOf(T_i) \neq PeriodOf(T_j)$.

## 3.3 Restoring Visual Details via Additional Visual-Semantic Supervision

Artifact images generated by the vanilla Stable Diffusion model suffer from blurry edges and false color and patterns under the current setting (see Figure 1), implying that stricter visual constraints need to be enforced to address these issues. Therefore, we propose to use **edge loss** [31] and **perceptual loss** [15] that apply additional visual-semantic supervision on images generated by our Stable Diffusion model.

***Edge Loss.*** Building upon the insights from [31], we penalize the differences in contours between two images by aiming to minimize the $L_2$ distance between their edge maps, as shown in part **c)** of Figure 3. Since the vanilla Stable Diffusion model often produces images that suffer from the problem of incorrect and blurry shape compared to the ground-truth artifact, it is necessary to penalize such errors as defined here in the edge loss:

$$L_{\text{edge}} := ||EDGE(I_i) - EDGE(I_i')||^2 \qquad (3)$$

where $EDGE(\cdot)$ is an edge extracting function. In our approach, we use the Canny Edge Detector [4] as our edge extractor to extract edge maps (see examples in Figure 4), then compare the difference between two contours in $L_2$ distance.

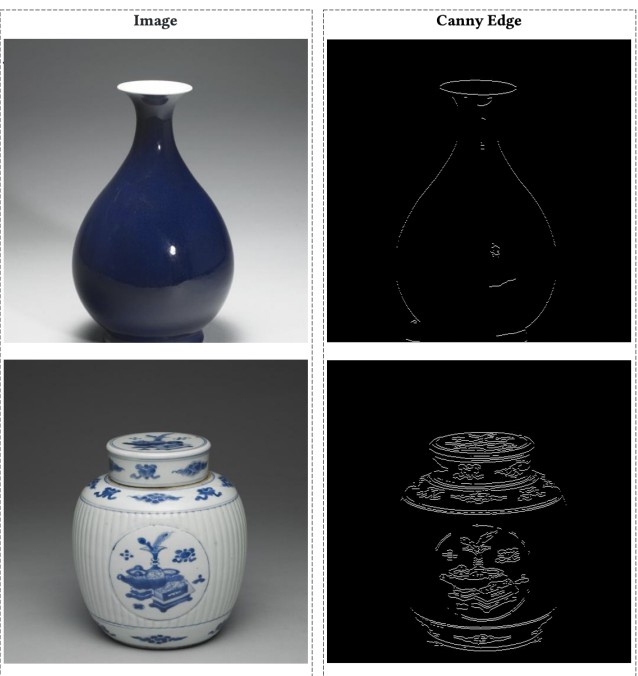

**Figure 4: Canny edge maps of artifacts**

**Perceptual Loss.** Similar to [15], we also penalize the problem of mismatching high-level details between the generated image and the real one. As we have also observed on the images generated by the vanilla Stable Diffusion model, the high-level details (such as colors and patterns) are often misaligned with the original ones. Therefore, we incorporate perceptual loss into our training process to tackle such an issue, as perceptual loss works by mapping the images into a semantic space using a pretrained network, and then minimizing the difference between the high-level features of the generated image and the original image. The formula for perceptual loss is defined as:

$$L_{\text{perceptual}} := ||\phi(I_i) - \phi(I_i')||^2 \qquad (4)$$

where $\phi$ denotes a pretrained image encoder to extract the high-level features of an image. This is applied to impose stricter supervision on color, texture, and other high-level features. In our method, we use a *CLIP-ViT-L/14* [25] image encoder to act as our pretrained image encoder for perceptual loss.

### 3.4 Objective Functions

Combining all the extra multi-source multi-modal supervisions above, the overall training objective of our system is:

$$L := L_{SD} + \lambda_1 \, L_{\text{text}}(x_i, X)$$
$$+ \lambda_2 \, L_{\text{edge}}(I_i, I_i') + \lambda_3 L_{\text{perceptual}}(I_i, I_i') \qquad (5)$$

where $\lambda_1$, $\lambda_2$ and $\lambda_3$ are hyperparameters controlling the weight of each supervision loss; $x_i$ is the similarity between a positive sample pair yielded from $T_i'$ and $X$ is a set of similarities of sampled negative pairs; $I_i$ and $I_i'$ are the ground-truth and the restored image from our model's prediction.

## 4 EXPERIMENTS

### 4.1 Experimental Setup

**Dataset.** Due to the sparsity of paired text-image data in the ancient artifact domain, we build our own text-to-image dataset by collecting artifact information from *National Palace Museum Open Data Platform* [41]. After careful cleansing of available entries, we are left with 16,092 unique artifact samples with their descriptions and ground-truth images. We split the data by 80%/10%/10% for training, validation, and testing. [3]

**Implementation Details.** For our backbone model, we use a pre-trained Chinese Stable Diffusion *Taiyi-Stable-Diffusion-1B-Chinese-v0.1* [51] (dubbed *Taiyi-SD*) which was trained on 20M filtered Chinese image-text pairs. *Taiyi-SD* inherits the same VAE and U-Net from *stable-diffusion-v1-4* [28] and trains a Chinese text encoder from *Taiyi-CLIP-RoBERTa-102M-ViT-L-Chinese* [52] to align Chinese prompts with the images. Further training details are left in Appendix C.

**Evaluation Metrics.** To comprehensively evaluate our method for text-to-image synthesis quantitatively, we employ three commonly used metrics that measure image generation quality: **CLIP Visual Similarity**, **Structural Similarity Index (SSIM)** [46] and **Learned Perceptual Image Patch Similarity (LPIPS)** [53]. Each of them highlights different aspects of the generated image. Together, they provide a thorough judgment of a synthesized artifact image in terms of its overall resemblance to the ground truth, the accuracy of its shape and pattern, and its perceptual affinity to the target image. We leave an extensive explanation of these metrics in Appendix D.

### 4.2 Main Results and Discussion

In Table 2, we compare the quantitative results of our approach with the baselines on our test set. The first column denotes the models we experimented with.

For the baselines, we use *Taiyi-SD* via two versions:

- **Taiyi-SD-finetuned-description:** the finetuned *Taiyi-SD* with the raw description (directly available from museum archives) as input prompt;
- **Taiyi-SD-finetuned-attributes:** the finetuned *Taiyi-SD* using LLM-enhanced prompt (a sequence of artifact attributes) as designed in Section 3.1.

For our approach, we apply the LLM-enhanced prompt by default and also explore three different versions of extra supervisions in addition to training the *Taiyi-SD* backbone:

---

[3]To facilitate further research, we will make the dataset as well as our ChatGPT-enhanced prompts publicly available.

**Table 2: Quantitative comparison of our models against the finetuned *Taiyi-SD* baselines over CLIP Visual Similarity (CLIP-VS), SSIM and LPIPS.**

| Models | Prompt | $\lambda_1$ | $\lambda_2$ | $\lambda_3$ | CLIP-VS ↑ | SSIM ↑ | LPIPS ↓ |
|---|---|---|---|---|---|---|---|
| *Taiyi-SD*-finetuned-description | raw description | - | - | - | 0.772 | 0.536 | 0.608 |
| *Taiyi-SD*-finetuned-attributes | LLM enhanced attribute | - | - | - | 0.792 | 0.554 | 0.598 |
| *OURS*-attributes +text | LLM enhanced attributes | 0.5 | - | - | 0.801 | 0.580 | 0.552 |
| *OURS*-attributes +edge+perceptual | LLM enhanced attributes | - | 0.3 | 0.1 | 0.815 | **0.636** | **0.497** |
| *OURS*-attributes +text+edge+perceptual | LLM enhanced attributes | 0.3 | 0.3 | 0.1 | **0.831** | 0.594 | 0.536 |

**Table 3: Quantitative comparison between zero-shot *Taiyi-SD* models using different textual prompts over CLIP Visual Similarity (CLIP-VS), SSIM and LPIPS.**

| Prompt | CLIP-VS ↑ | SSIM ↑ | LPIPS ↓ |
|---|---|---|---|
| Raw description | 0.748 | 0.383 | 0.748 |
| LLM enhanced attributes-sequence | **0.765** | **0.413** | **0.730** |

**Table 4: Human evaluation of the quality of artifact images generated by the finetuned baseline and our model. The images are rated from 5 different aspects (see Section 4.4) on a scale of 0 to 5 by 20 archaeology experts from top institutions.**

| Models | Material ↑ | Shape ↑ | Pattern/Color ↑ | Size/Ratio ↑ | Dynasty ↑ | total avg. ↑ |
|---|---|---|---|---|---|---|
| *Taiyi-SD*-finetuned | 2.66 | 1.50 | 1.44 | 1.79 | 2.12 | 1.90 |
| *OURS* | **3.94** | **3.38** | **3.25** | **3.30** | **3.20** | **3.41** |

- ***OURS*-attributes +text:** finetuning with additional text contrastive loss (see Section 3.2) to align the text representation of our model with domain expertise;
- ***OURS*-attributes +edge+perceptual:** finetuning with edge loss and perceptual loss (see Section 3.3) as additional supervision to enforce more visual-semantic constraints on the image generation process;
- ***OURS*-attributes +text+edge+perceptual:** finetuning with both text contrastive loss and the edge and perceptual loss as multi-source supervision.

Overall, our proposed artifact image synthesis approach significantly outperforms the finetuned *Taiyi-SD*-baselines across all metrics. The improvement on SSIM indicates that images generated by our model better preserve the shapes and boundaries of the original artifacts. An increase in CLIP Visual Similarity also indicates that our approach produces images that are more closely aligned to the ground truths.

Additional visual-semantic constraints in the form of edge loss and perceptual loss contribute greatly to boosting the SSIM and LPIPS scores. This can be attributed to the fact that edge loss and perceptual loss put a stricter condition on both structural details like **edge and contour** (captured by **SSIM**) and perceptual-level image features like **color and texture** (captured by **LPIPS**). These visual details are exactly much desired in our case of artifact image synthesis, as the shape, pattern, and texture of artifacts are of vital importance for determining their historical position and status.

By further incorporating the text contrastive loss into the overall training objective, we observe a slight increase in CLIP Visual Similarity, yet a decrease in SSIM and LPIPS scores. We believe

there are two reasons behind this phenomenon. For one, by aligning the text knowledge (descriptions with names) (see Section 3.2), the textual guidance for generating the image is better represented and closer to the general visual content of the artifact, thus leading to a higher CLIP Visual Similarity. For another, the relative weight of edge and perceptual loss is reduced with the additional text contrastive loss, which might compromise the strict supervision on structural coherence and perceptual similarity and limit the model's performance on SSIM and LPIPS.

As is also evidently shown by just comparing the baselines finetuned with different prompt formats in Table 2, using LLM to enhance the prompt construction as a sequence of important artifact attributes effectively improves the performance of the finetuned baseline model across all three metrics. More about the effects of our LLM-enhanced prompting method will be discussed in the following subsection 4.3.

## 4.3 Ablation Studies

To investigate the contribution of components proposed in our approach and for further studies, we conduct extensive ablation studies on two key designs of our model.

***Effectiveness of LLM-enhanced prompts.*** As is shown in Table 2, the finetuned model benefits from LLM-enhanced prompting, achieving better scores on all three quantitative metrics. To further illustrate the effectiveness of our proposed prompting method, we explore the **zero-shot** setting, where the baseline *Taiyi-SD* is directly prompted to generate artifact images without any training on our artifact dataset. We use either the raw description from the

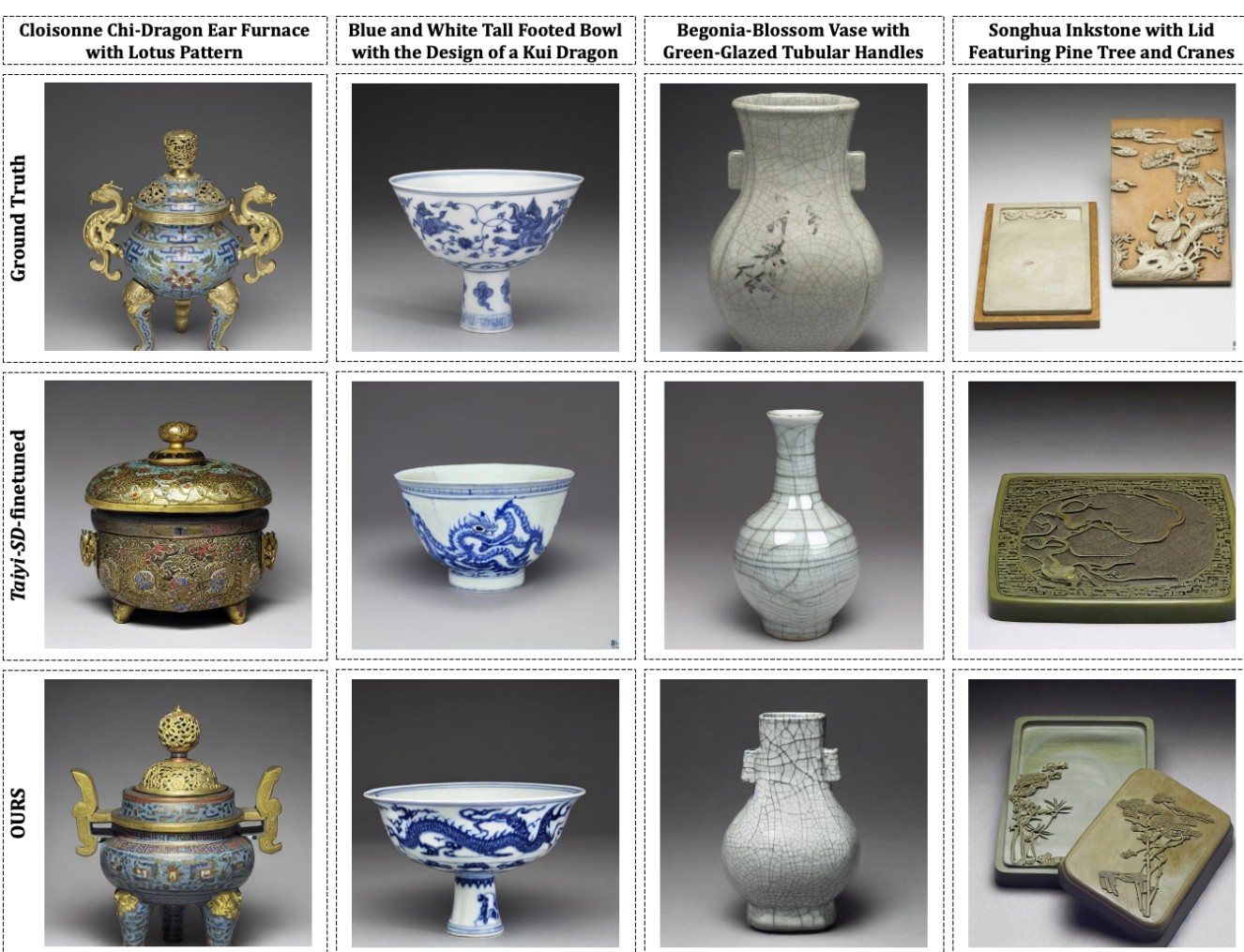

**Figure 5: Comparison between the finetuned *Taiyi-SD* baseline model and *OUR* approach trained with additional edge loss and perceptual loss against the ground truth. Clearly, objects generated by *OUR* model display more accurate shapes, colors, and patterns when compared to the ground truth, whereas these delicate visual details are easily neglected by the baseline.**

museum archives or the sequence of artifact attributes enhanced by LLM as prompt. The results, as shown in Table 3, again demonstrate the superiority of LLM-enhanced prompting, which excels across all metrics. This can be credited to the organized information format in the attribute sequence and the additional knowledge provided by LLM (see Section 3.1).

***Effectiveness of Edge Loss and Perceptual Loss.*** In Figure 5, we compare the artifact images generated from our model that uses edge loss and perceptual loss against the finetuned *Taiyi-SD* baseline that does not involve these visual semantic constraints. Evidently, the shapes, colors, and patterns of the artifacts are more accurate and close to the ground truth if the model is additionally supervised by edge and perceptual loss. On the other hand, the vanilla finetuning paradigm may easily lead to output objects either lacking proper shapes and forms or manifesting incorrect motifs and patterns. For example, in the second column of Figure 5, the

"tall-footed" aspect of the target bowl is clearly neglected without edge and perceptual constraints. Also, the intricate cracking lines on the "begonia-blossom shaped vase" shown in the third column are better simulated with our model.

## 4.4 User Study

In addition to quantitative evaluation, we conducted a user study involving archaeology experts to evaluate the generated images. This study is designed to assess various aspects of the generated artifacts, as outlined in our prompt design (see Section 3.1):

- ***Material:*** How accurately does the generated artifact resemble the actual manufacturing material?
- ***Shape:*** How closely does the generated artifact match the described shape?
- ***Pattern/Color:*** How faithful is the representation of patterns and colors on the generated artifact?

- **Size/Ratio:** How accurately does the generated artifact maintain the ratio of height and width?
- **Dynasty (Time Period)**: How well does the generated artifact reflect the characteristics of its era?

Each aspect is rated on a scale of **0** to **5**, with higher ratings indicating better quality. We randomly select 30 samples from the test set and provide the model-generated images to 20 graduate students of archaeology major from top institutions for assessment. The average ratings of images generated by our proposed method (*OURS*) are compared with those generated by the baseline Chinese SD model also finetuned on our data. The results are presented in Table 4.

Clearly, according to human experts, the artifact images generated by our method are much better in quality across all five important rating aspects, especially in terms of shape, pattern, and color. These results of our user study resonate with the findings from the automatic evaluation metrics and further highlight the superior performance of our model in generating artifact images that accurately align with history.

To offer a richer **qualitative demonstration** of our model's capabilities, we present a diverse collection of artifact images generated by our model, showcasing its remarkable fidelity across a broad spectrum of historical artifacts. Refer to Figure 6 in Appendix E [4] for a comprehensive visual display.

## 5 RELATED WORK

***Multimodal AI for Cultural Preservation.*** In the domain of fostering the preservation of art and culture, multimodal learning technologies have had an increasing presence in recent years. Revolving around ancient artworks (especially paintings with rich cultural backgrounds), datasets have been released that associate artworks with textual descriptions or question-answer pairs [1, 8, 40], which in turn inspires algorithmic explorations for the visual question answering (VQA) task on these artworks [2, 54]. In terms of historical artifacts, most recent works focus on restoring ancient character images. These include degraded images of characters printed in old documents [33, 34] as well as inscriptions on oracle bones [5]. [32] also adapts image captioning techniques to generate descriptions of ancient artifacts sourced from museums.

With a similar objective to aid cultural preservation, our work - as pointed out in Section 1 - represents the first novel approach to exploring text-to-image reconstruction of physically lost ancient artifacts that only exist in textual documentation. Our proposed system now enables archaeologists to easily obtain a highly accurate visual simulation of any artifact of interest mentioned in historical texts, thus opening up new possibilities for research into our cultural heritage and detailed artifact analyses.

***Text-to-Image Synthesis.*** Text-to-image synthesis tasks have long been a vital task at the intersection between computer vision and natural language processing, of which models are given a plain text description to generate the corresponding image. One major architecture in this area is GAN [9], whose variations [16, 48, 50] have resulted in the state-of-art performance of text-to-image synthesis tasks. Recently, diffusion models [14, 27, 35, 37, 38] also have

demonstrated their ability to achieve new state-of-the-art results [6]. Diffusion models make use of two Markov chains: forward and reverse. The forward chain gradually adds noise to the data with Gaussian prior. The reverse chain aims to denoise the data gradually. The transition probability at each timestep is learned by a deep neural network, which in the case of text-to-image synthesis is usually a U-Net [29] model.

***Large Language Models.*** Language models are a family of probabilistic models that predict the probability of the next word, given a sequence of previous words within a context. The introduction of GPT-3 [3], which contains 175B parameters, has led to the emergence of Large Language Models (LLMs), referring to language models with a large number of parameters. These LLMs have demonstrated never-seen-before abilities, expanding the frontiers of what is possible with language models. One emerging ability of LLMs is in-context learning [3], where LLMs are able to perform downstream tasks after being prompted with just a few examples without further parameter updates. Thus, by providing carefully designed examples, we can make use of LLMs as an information extractor given a noisy and unstructured text. LLMs have also shown their ability to acquire world knowledge from the massive training corpus [21, 24, 44]. An efficient way to extract the implicit knowledge from LLMs is to ask questions with proper prompt engineering as LLMs are highly sensitive to the prompt input [20].

## 6 CONCLUSION

In this paper, we present a novel approach to tackle the challenge of artifact image synthesis. Our method features three key techniques: **1)** Leveraging an LLM to infuse textual prompts with archaeological knowledge, **2)** Aligning textual representations with domain expertise via contrasting learning, and **3)** Employing stricter visual-semantic constraints (edge and perceptual) to generate images with higher fidelity to visual details of historical artifacts. Quantitative experiments and the human evaluation from our user study with archaeology experts confirm the superior performance of our approach compared to existing models, significantly advancing the quality of generated artifact images.

Beyond technological contributions, our work introduces a more profound **societal impact**. As the first attempt to restore lost artifacts from the remaining textual descriptions, our work empowers archaeologists and historians with a tool to resurrect lost artifacts visually, offering new perspectives on cultural heritage and enriching our understanding of history. We also hope that this work will open new avenues for further exploration, fostering deeper insights into our past and cultural legacy with the help of technical advances.

## 7 ETHICS STATEMENT

In this project, we only used training data sourced from *National Palace Museum* [41]. This ensures that the data we worked with has already been scrutinized by authorities and is open to the public. However, we recognize possible inaccuracies in our model's generation despite our extensive efforts to improve its fidelity. Therefore, anyone using our model and system should be warned of possible mistakes in the generated artifact images and we strongly advise all users to verify important content.

---

[4]All **Appendices** are submitted as supplementary material.

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
