# OpenReview forum: "Knowledge-Aware Artifact Image Synthesis with LLM-Enhanced Prompting and Multi-Source Supervision"
_acmmm.org/ACMMM/2024/Conference — MM2024 Poster_

### Official Review · Reviewer_Vosy · 2024-05-14

**Rating:** 2
**Confidence:** 3

**Summary:**

This paper presents a novel approach to tackle the challenge of artifact image synthesis. It adopts a large language model to enrich the textual description with archaeological knowledge. A contrastive learning strategy is used to align the object names and their descriptions. Then the edge loss and perceptual loss are adopted to improve image fidelity. Quantitative experiments and the user study with
archaeology experts verify the strength of this method compared to the simple finetuning method.

**Strengths:**

This paper studies a novel problem of artifact image synthesis, which may benefit archaeological study.

The knowledge of LLM is imported to enhance the synthesis of images in a specific domain.

Some visual results and the user study exhibit the superiority of the proposed method.

**Limitations:**

The training process is not clear, and there could be technical mistakes:

-- Is the Canny detector differentiable?

-- How to obtain I_i^' during training? Did the author update the VAE parameters?

-- Was the denoising loss backpropagated to the text encoder?

-- How to use the validation set?

The evaluation:

-- How many images were sampled for each textual description? For this task, do the authors expect the diversity of synthesized images?

-- The qualitative comparison is insufficient.

-- Did the author try to synthesize some non-existent objects? Since it is one of the possible applications of this model.

-- Incorporating the text contrastive loss improves the CLIP similarity but decreases the SSIM value. It would be better to provide more visual results, and a user study could evaluate these two versions.

Other suggestions:

-- The textual descriptions are not provided in Figure 1.

-- Figure 4 seems unnecessary.

**Suitability:**

3

---

### Official Review · Reviewer_v1TG · 2024-05-23

**Rating:** 5
**Confidence:** 3

**Summary:**

The paper addresses the challenge of recreating visual images of ancient artifacts from textual descriptions, given the limited availability of physical copies. It highlights the significance of this task for historical studies and cultural preservation. Despite recent advancements in text-to-image synthesis using diffusion models, the paper identifies shortcomings in applying these models to archaeological studies due to the lack of domain-specific knowledge supervision during the generation process. To overcome these challenges, the paper proposes a knowledge-aware artifact image synthesis approach, utilizing Large Language Models (LLMs) to enhance text prompts and introducing additional supervision mechanisms for both textual and visual modalities. Experimental results and a user study demonstrate the effectiveness of the proposed approach in improving the generation quality of historical artifacts, showcasing its potential to aid archaeologists in gaining deeper insights into cultural heritage.

**Strengths:**

1. The paper introduces a novel approach to address the specific challenge of recreating visual images of ancient artifacts from textual descriptions, contributing to the intersection of artificial intelligence and archaeology.

2. By integrating Large Language Models and introducing multimodal supervision mechanisms, the paper proposes a comprehensive solution tailored to the specialized domain of archaeological studies, addressing the limitations of existing text-to-image synthesis methods.

3. The paper provides rigorous experimental evaluation and user studies, demonstrating the superior performance of the proposed approach compared to existing models, thereby validating its effectiveness and potential utility for researchers and practitioners in the field.

**Limitations:**

1. FID serves as a vital metric for assessing the quality and diversity of generated models. The authors should incorporate FID as a quantitative measure and compare it with baseline models.
2. The use of Large Language Models (LLMs) to assist in generating models in the visual domain has been explored extensively. The authors should discuss the distinctions from existing methods and highlight the unique aspects of employing LLMs in this paper.
3. The rationale behind utilizing Chinese Stable Diffusion as the baseline model needs clarification. Official Stable Diffusion models are expected to possess superior generation capabilities. The authors should discuss the advantages of using Taiyi-Stable-Diffusion.

**Suitability:**

3

---

### Official Review · Reviewer_RAfn · 2024-05-24

**Rating:** 5
**Confidence:** 4

**Summary:**

This paper introduces a knowledge-aware approach for synthesizing images of ancient artifacts, effectively capturing their visual forms by integrating archaeological knowledge, historical expertise, and visual-semantic constraints. Compared to existing methods, the proposed approach generates higher-quality images that better align with the implicit details and historical knowledge from textual descriptions, showcasing significant improvements over prior arts.

**Strengths:**

1. Recreation of visual images of ancient artifacts from text descriptions is a novel and intriguing research direction, and this manuscript makes solid explorations in this area.
2. The authors propose a novel method for prompt construction that leverages domain knowledge, which is highly meaningful for the research community facing data scarcity.
3. The manuscript is written clearly and is easy to follow.

**Limitations:**

1. The generated images of artifacts lack detail. For instance, in the first column of Fig. 5, where the model recreates the "Cloisonne Chi-Dragon Ear Furnace with Lotus Pattern," while the proposed method captures the artifact's overall shape, the "ear" part notably deviates from the ground truth. Given the historical and cultural significance of these details, further research is recommended to enhance the generated artifact images' level of detail.
2. There is a lack of a quantified ablation study. Despite providing visual ablations in Section 4.3, the manuscript lacks a quantitative evaluation of each component's effects. Incorporating a quantified ablation study would enhance the manuscript's quality.
3. The manuscript employs a base model based on Stable Diffusion v1.4, as mentioned in Section 4.1. If a more advanced base model were utilized, such as SD-XL [1], would it enhance the proposed method's performance? Further discussion on this point would be beneficial.

[1] Podell, Dustin, et al. "Sdxl: Improving latent diffusion models for high-resolution image synthesis." arXiv preprint arXiv:2307.01952 (2023).

**Suitability:**

3

---

### Official Review · Reviewer_Fk9J · 2024-06-01

**Rating:** 5
**Confidence:** 2

**Summary:**

The paper introduces a novel approach for re-visualizing ancient artifacts using text-to-image synthesis. By incorporating explicit archaeological knowledge, additional textual guidance related to historical expertise, and visual-semantic constraints, the proposed model generates higher-quality artifact images that align better with historical details. This research contributes to preserving cultural heritage and commemorating our past through reconstructed visual forms based on textual descriptions.

**Strengths:**

the paper found good point, namely the generation and restoration of artifacts. The two issues raised in the paper are indeed challenges faced by current large language models and vision-language multimodal models. The author enhances semantic understanding by utilizing enriched prompts, optimizes training strategies, and strengthens vision-language modality comprehension by increasing the loss function.

**Limitations:**

The description of the paper is quite clear. However, I have a concern.

In Table 2, the last two rows show that after adding text information, the SSIM and LPIPS metrics worsened. Can this be interpreted as a degradation in visual capability? Why does this phenomenon occur? The paper does not provide an analysis of this issue.

**Suitability:**

3

---

### Meta-Review · Area_Chair_gYhr · 2024-07-02

**Recommendation:** Accept (Poster)
**Confidence:** 4

**Metareview:**

This paper has received Weak Accept, two Borderline Accept, and one Borderline Reject. This paper has studied a very interesting research direction of the recreation of visual images of ancient artifacts. The paper is clearly written and easy to follow.The technical solution utilizes LLM to explore domain knowledge to enhance the performance. The authors are suggested to revise the description of the edge loss, as raised by one reviewer. FID is also suggested to be added for completeness. Thus, the AC recommends the acceptance of the paper.